# Blue Light—Ocular and Systemic Damaging Effects: A Narrative Review

**DOI:** 10.3390/ijms24065998

**Published:** 2023-03-22

**Authors:** Răzvan-Geo Antemie, Ovidiu Ciprian Samoilă, Simona Valeria Clichici

**Affiliations:** 1Department of Physiology, Faculty of Medicine, “Iuliu Haţieganu” University of Medicine and Pharmacy, 400006 Cluj-Napoca, Romania; 2Department of Ophthalmology, Faculty of Medicine, “Iuliu Hatieganu” University of Medicine and Pharmacy, 400006 Cluj-Napoca, Romania

**Keywords:** blue light, retina, age-related macular degeneration, intrinsically photosensitive retinal ganglion cells, intraocular lens, LED, photophobia, migraine

## Abstract

Light is a fundamental aspect of our lives, being involved in the regulation of numerous processes in our body. While blue light has always existed in nature, with the ever-growing number of electronic devices that make use of short wavelength (blue) light, the human retina has seen increased exposure to it. Because it is at the high-energy end of the visible spectrum, many authors have investigated the theoretical harmful effects that it poses to the human retina and, more recently, the human body, given the discovery and characterization of the intrinsically photosensitive retinal ganglion cells. Many approaches have been explored, with the focus shifting throughout the years from examining classic ophthalmological parameters, such as visual acuity, and contrast sensitivity to more complex ones seen on electrophysiological assays and optical coherence tomographies. The current study aims to gather the most recent relevant data, reveal encountered pitfalls, and suggest future directions for studies regarding local and/or systemic effects of blue light retinal exposures.

## 1. Introduction

The retina consists of a plethora of cells and connections that form ten distinct layers, which are capable of receiving and interpreting electromagnetic radiation in the range of roughly 400 nm (blue)–700 nm (red) [1]. Its inner part consists of the retinal nerve fiber layer formed by the axons of the lower retinal ganglion cells layer. They form the optic nerve through which the retina conveys signals to different parts of the brain. Its outermost layer—the retinal pigment epithelium (RPE)—has a variety of roles in the human eye: regulation of retina development [2], reducing photo-oxidative stress, secretion of growth factors, transportation of metabolites and fluids (as part of the outer blood–retinal barrier), and phagocytosis of used rod and cone outer segments [3]. It plays the crucial role of supplying vital nutrients, creating the healthy cellular milieu necessary for the proper functioning of the cones and rods layer (photoreceptor layer). Blue light, at the high-energy end of the visible spectrum, may represent a risk for retinal damage [4]. Prolonged exposure increases the number of reactive oxygen species (ROS) and promotes lipofuscin deposition in RPE cells, drusen, and choroidal microvascular changes, all contributing to a greater increase in age-related macular degeneration (AMD) in patients.

AMD is a multifactorial eye disease that takes years to progress, making it hard to directly assess in vivo the destructive effects of blue light. It is known that age, smoking, nutritional status, [5] sunlight exposure, and genetic background are risk factors. The fact that the retinoid A2E (N-Retinylidene-N-retinylethanolamine, the major fluorophore of lipofuscin) accumulates as aging occurs constitutes an even greater risk for cell apoptosis [6,7,8,9], as this retinoid is highly responsive to high-energy blue radiation. There are many recent attempts to block blue light phototoxicity, ranging from spectacle lenses to intraocular lenses (IOLs) and even a dietary approach, but all with variable and controversial benefits [5,10,11,12]. 

The classic view considers the cones and rods as the main receptors of light. However, in the last 20 years, there has been an increased interest in the inner retinal layer, more precisely in retinal ganglion cells, as a set of observations and experiments have raised the possibility of another type of cell being responsive to light [13]. In 2002, Samar et al. managed to pinpoint the structure responsible for this—melanopsin [14]. This pigment, which has its peak absorption in humans at 480 nm (blue light), was found to exist inside only a 1–2% subpopulation of retinal ganglion cells [14,15,16,17,18]. Because of its property to respond to light and to generate influx independently from cones and rods, this type of cell has been named the intrinsically photosensitive retinal ganglion cell (ipRGC) [13]. It is now recognized that blue light stimulation of the ipRGC plays an important part in the non-image-forming response to light, being its central mediator [19,20,21]. Thus, it became clear that the retina acts and responds to light, mainly blue light, in different ways because of the two systems that it incorporates: the image and non-image visual response systems.

Photophobia, a sensation of visual discomfort due to light commonly found in neurological and ophthalmic disorders, is sometimes present in healthy individuals at a light level that would otherwise be perceived as being pleasant by most. Patients with corneal abrasions, iritis [22], and trigeminal neuralgia [23] frequently experience photophobia as well as patients with migraines (migraineurs) during their ictal [23] and interictal periods [24]. The mechanism for this process is not yet fully understood. Many theories have been proposed, starting with Hopkinson’s pupillary hippus [25], which suggested that the iris has a role in the pain-signaling mechanism for visual discomfort. Years later, Okamoto et al. [26], in their experiments with albino rats, demonstrated the role that the olivary pretectal nucleus (OPN), an important station in the pupillary control circuit, plays in photophobia. Photophobia is thus mediated by ipRGCs and trigeminal afferents [27,28]. Electrophysiological recordings of trigeminovascular neurons in the thalamus show that inputs from ipRGCs increase their firing rate. Additionally, axons from these dura-sensitive neurons project to cortical areas such as the primary somatosensory and to regions of the primary and secondary visual cortices, possibly explaining the light–photophobia–migraine link [28]. Other studies also indicate that nociceptive brainstem neurons belonging to the trigeminal pain-signaling pathway seem to be activated by signals fired from the OPNs which in turn receive inputs from the ipRGCs [13,15,29,30]. Thus, pupil dynamic may be seen as an indicator of ipRGC activation, adding a new dimension to the role of the iris in photophobia.

At least one group (Stringham et al. [31]) determined that macular pigment (MP) has an important role in photophobia and that ipRGCs are part of an important risk-avoidance system designed to protect the fovea by eliciting this behavioral response [32]. Studies with blind migraineurs that still experience light avoidance during ictal episodes when compared to those who have undergone enucleation [28] are a strong line of evidence for the implications of ipRGCs at a systemic level. 

Over the past decade, a growing number of studies have focused on the effects of blue light on the retina and, more recently, on the human body as a whole. This is especially relevant given the high use of smartphones and tablet devices [33,34,35,36] that use a solid-state lighting technology that emits a higher amount of blue light despite its white light appearance [35,36]. Blue light plays a significant role in our day-to-day lives, penetrating deeply into education, modern industries that run 24/7, and most importantly, our lifestyle, to such an extent that it seems impractical and almost impossible to avoid exposure to it. Thus, the present narrative review aims to offer a wider perspective of the harmful biological consequences after the human retina is exposed to blue light, gathering and interpreting the effects both at a local and systemic level. Effects regarding photoentrainment, melatonin and other hormone secretion, sleep quality and next morning alertness, axial length, and refractive status of the eye concerning blue light have not been taken into consideration. 

## 2. Methods

This manuscript was prepared following the indications and procedures provided by Green et al. [37], Grant et al. [38], and Pautasso [39], as well as following the guidelines described by Baethge et al. [40] regarding the SANRA (Scale for the Assessment of Narrative Review Articles). Studies between 2010 and 2022 were searched in the following databases: PubMed/Medline, Scopus, EMBASE, Cochrane Database of Systematic Reviews, and Clinical Trials. The search strategy included the following keywords: “blue light”, “retina”, and “intrinsically photosensitive retinal ganglion cell”.

Regarding eligibility, clinical studies and randomized controlled trials reported in English with human participants were selected. Studies with fewer than 10 human participants were excluded from this review.

## 3. Results

Two distinct searches were performed to obtain a comprehensive image of the subject. The first one included the keywords “blue light” and “retina”, focusing on the damage assessment of rods and cones when exposed to blue light, while the second one included “intrinsically photosensitive retinal ganglion cell”, searching for the harmful effects of these cells on the body. After applying the filters (Year: “2010 to 2022”; Article type: “Clinical Study”, “Clinical Trial”; Species: “Human”; Language: “English”) to search the above-mentioned databases, a total of 333 studies remained. After reading their abstracts and applying the exclusion criteria, 20 abstracts were selected. After fully reading the articles, three were eliminated because the number of participants was less than 10 per study (Figure 1). Table 1, Table 2 and Table 3 summarize the main characteristics of each of the 17 studies included in the final review.

## 4. Discussion

Previously, researchers have centered their attention mainly on the effects of blue light exposure on the retina, but recent data also highlight systemic implications. The research groups of the selected seventeen studies had different approaches to this common theme—eight of them assessed retinal (macular) health using either different types of IOLs (five studies) or employing electrophysiological assays (three studies). The other nine explored the blue light–photophobia–migraine link, the autonomic nervous system, or psychomotor function, thus integrating systemic consequences with this type of light exposure. 

### 4.1. IOLs and Their Protective Role

Currently, there is no clear consensus on whether or how blue light plays a role in the pathogenesis of AMD [41,57]. In the review by Mainster et al. [58], the authors highlight the idea that most AMD cases appear in phakic adults over 60 years old and, despite the photoprotection conferred by the lens, which is greater than that of a blue light-filtering IOL, AMD still does not cease to appear.

The beginning of the new millennium has seen a change in the type of IOLs being implanted in a patient’s eye—from the classic clear ultraviolet light-filtering IOLs to the yellow-tinted blue light-filtering IOLs. The reason behind it is that the latter prevents the blue light exposure of the retina, especially abundant in A2E with aging, and as such, prevents DNA damage and cell apoptosis mediated by singlet oxygen, which is wavelength dependent [59,60,61,62]. Studies performed by Sparrow et al. and Yanagi et al. [60,63] showed the inhibition of vascular endothelial growth factor (VEGF) production as well as a protective role of the blue-blocking IOL on RPE cells, while Obana et al. [64], by performing Raman spectroscopy, highlighted a reduction in macular pigment optical density (MPOD) in the clear IOL group. Other authors explored the effects of blue light with various tests: visual acuity (VA), contrast sensitivity (CS), color discrimination, glare, photopic/scotopic/mesopic sensitivity, ocular coherence tomography (OCT), fundus autofluorescence (FAF), and ophthalmoscopy. 

Neumaier-Ammerer et al. were the first to test four types of IOLs (two yellow-tinted and two clear) made by different manufacturers. In their various light intensity settings (10, 100, and 1000 lx), the authors found significant differences regarding the tritan axis only under mesopic conditions, with the yellow-tinted IOL group making more mistakes in the blue light-spectrum in the first and the eighth week postoperatively than the clear IOL group. Other differences between the two tinted IOL study groups regarding VA, CS, and glare were not detected [41]. Despite using the Roth 28 Hue Test (28 disks), which is not as sensitive as the Farnsworth-Munsell 100 Hue Test (85 disks), the group still found a statistical difference when color vision was evaluated, though only in mesopic conditions.

The benefits of tinted IOLs were also evaluated by Kara Junior et al. [42] in patients with bilateral cataract surgery, with one eye randomly receiving a tinted IOL and the other eye receiving a clear IOL one week apart. Color vision under photopic conditions, contrast sensitivity under both photopic and scotopic conditions, and macular findings (utilizing ophthalmoscopy and OCT) concerning AMD five years after surgery were the primary endpoints of the study. The authors failed to detect a difference between the study groups when central macular thickness, contrast sensitivity, and color vision were assessed. Using a randomized intraindividual design allowed them to avoid variables that could potentially affect data, such as a diet with a high intake of antioxidants and minerals, which is known to impact MPOD [5,65].

Because abnormal FAF can be an early predictive sign for AMD, Nagai et al. [43] assessed its changes two years after cataract surgery between a tinted and a clear IOL model using the Heidelberg Retina Angiograph 2. In this prospective observational study, the authors evaluated and compared the development, progression, or decrease in abnormal FAF, drusen, and the development of wet AMD (wAMD) or geographic atrophy (GA). The yellow IOL group had a statistically significant lower incidence of any form of AMD and abnormal FAF findings, whereas the drusen progression in the clear IOL group (3.8%) did not reach statistical significance. After categorizing FAF abnormalities into eight domains, it was shown that the reticular pattern dominated (21/34 eyes with abnormal FAF) and that the patchy pattern was the one linked with the highest risk of AMD progression (4/5 eyes with patchy pattern). FAF can be used to search for early signs of AMD, and even though Nagai et al.’s study had a small number of enrolled patients and a non-randomized design, it brought new arguments in favor of the idea that yellow-tinted IOLs might potentially protect the retina against post-cataract surgery blue light hazard.

Ayaki et al. wanted to observe the differences between clear and blue-blocking IOLs, after performing cataract surgery and increasing retinal photoreception, with the use of the Japanese-adapted version of the National Eye Institute Visual Function Questionnaire (VFQ-25) and the Pittsburg Sleep Quality Index (PSQI) [44]. One of the sub-scales of the VFQ-25 is ocular pain, and the authors observed that the greatest change was seen in the group which received a blue-blocking IOL, thus linking more blue light exposure of the retina with more ocular pain. Other differences between the study groups regarding sub-scales of the VFQ-25, such as color vision, distance, and near vision, were not of statistical significance. When examined more carefully, some comments should be made. The authors obtained a paradoxical postoperative result regarding ocular pain—an improvement, which was best seen in the yellow-tinted IOL. Pain is not well defined in this context, and it is plausible that the patients may have interpreted the reduction in preoperatively photophobia/glare as a reduction of this parameter. This could explain the better results that the authors obtained in the yellow-tinted IOL group, and we can judge it as an indirect marker of less ipRGC stimulation, as it plays an important role in photophobia. Although their study had an impressive number of 206 participants, the short study period, and the lack of IOL randomization and an objective way of assessing ocular function (such as using OCT, ERG, or pupillometry) make it hard to derive definite conclusions concerning the protective effects of the blue-blocking vs. clear IOL and, indirectly, of the effects of blue light on visual function.

Color vision was also an area of interest for Mokuno et al. [45]. Patients with macular pathology and controls were implanted with a blue-blocking IOL and later tested using the Farnsworth-Munsell 100 Hue Test under different lighting conditions. Patients were tested for at least 20 days and the results showed that the total error scores were not significantly different between the macular and non-macular disease groups under both photopic and scotopic conditions. Even though color vision was the main outcome, the lack of difference between the two groups can be seen as an indirect marker showing that blue light did not affect an already-diseased retina.

### 4.2. Retinal Electrophysiological Responses to Blue Light

Gagné et al. [46] were the first to show that short exposures to blue light can impact both cone and rod ERG responses even when exposure was within the standard safety range. The b-wave decrease was observed in both scotopic and photopic conditions with a similar magnitude. Reduced responses were still observable one hour after the blue light exposure ended. It was concluded that this reduction was only temporary. Although a selective blue light b-wave reduction was observed, with its reductions probably being caused by a decreased Müller cell potassic conductance caused by blue light, the group could not derive clear conclusions regarding the hazardous nature of blue light, their data adding to the still uncertain effects of this short wavelength on the human retina.

VDTs (mobile phones, computers, and others) may be a cause of various visual problems and have been seen to have a negative impact on the visual system [66,67,68], such as eye fatigue, which can be objectively evaluated with the use of CFF. Morita et al. [47] observed that CFF was lower after a two-hour acute VDT load on chronically VDT-exposed (>6 h/day for more than one year) healthy participants. By testing the effects of *Lactobacillus paracasei* KW3110 (*L. paracasei* KW3110) in vitro on human peripheral blood mononuclear cell-derived M2 macrophages and afterward on ARPE-19 cells, the group reported that this type of lactic acid bacteria had reduced the cell death rate most probably through the effects of IL-10. In the clinical setting, its administration alleviated both objective and subjective parameters, but these were significantly improved only four weeks after oral intake of *L. paracasei* KW3110 capsules. One must bear in mind the learning effect that might arise after repeated eye-fatigue evaluations. This could explain why the authors found no difference between the *L. paracasei* KW31110 group and the placebo group eight weeks after oral intake. Caution must be taken when interpreting these results.

Focusing their attention on VDTs as well, Li et al. conducted a clinical observational study in which they observed the chronic photodamage induced by the low-intensity blue light of phones [48]. With the use of mfERG, they examined the response of several small retinal regions in the macular area of 25 healthy participants whose use of VDTs was high. Those in the observational group (use of VDT > 8 h/day) displayed reduced amplitude of the outer retina in the parafoveal region of the macula as well as delayed peak time. Many recent studies observe acute retinal light damage [69,70,71], while chronic damage to blue light, being difficult to detect, is often overlooked, especially in low-light settings. However, Li et al. focused their attention on the long-term low-illuminance effects and concluded that usage of more than eight hours per day for more than five years induced parafoveal functional damage as shown by the mfERG results. The authors further investigated these effects on SD rats and found that chronic retinal photodamage involved all layers of the retina (cones, rods, bipolar cells, Müller cells, ganglion cells); it was cumulative and time-dependent. Thus, photoreceptor death is an important consequence of blue light damage to the retina. Because it resembles today’s modern way of living, the study design used by Li et al. comes as close as possible to the real degree of blue light damage to human eyes, highlighting its toxic effects on eye health.

### 4.3. Systemic Effects of Blue Light

#### 4.3.1. Photophobia and Migraine

A feature of the day-to-day life of people experiencing migraine episodes is photophobia [72,73,74,75], which is similar in those with and without visual aura [24,73,76,77]. Non-incandescent artificial indoor light has been seen to trigger headaches [78]. Noseda et al. and Digre et al. [27,28] have called attention to the connection between the ipRGC’s activation and the trigeminal nociceptive pathway, reinforcing the idea that migraine pain arises also from the eye [79], their results being consistent with those of Cajochen et al. [80] and Chellapa et al. [81]. Signals emanating from the retina and the trigeminal pathway have been shown to interact and potentiate each other, as demonstrated by the cases of migraine patients, where noxious trigeminal stimulation increases light sensitivity and enhances visual cortex activity [82,83], as well as in cases where light decreases pain thresholds of the trigeminal pathway activity [84], a phenomenon explained by the convergence of ipRGCs and trigeminal pathway on the posterior thalamus. Melanopsin expression in different human and mice trigeminal branches, especially in the ophthalmic one, could also contribute to photophobia [85,86]. This function is still debatable as several authors did not find a significant effect [54,87].

Besides IOLs, other ophthalmic devices have been developed to limit blue light exposure, such as spectacle lenses. With today’s increase in blue-enriched light-emitting diode (LED) backlight VDTs, our eyes are exposed more than ever. This type of spectacle lens claims to use filtering materials or surface coating to eliminate as much as possible the transmittance of this short wavelength and, as such, could to some extent represent a possible treatment for blue light-related disorders [10]. One notable trial was the one performed by Good et al. in 1991 [88] where, by testing FL-41 tinted spectacle lenses, the group found an almost four-times reduction in migraine frequency in a cohort of children. 

Modern technologies, such as the thin film combined with the optical notch filter used in Hoggan et al.’s study [49], have provided some interesting insights into ipRGCs and their role in photophobia and migraine. The study aimed to determine if the proposed filter could reduce headache impact in a cohort of chronic migraine patients using two thin-film optical notch pairs of spectacle lenses (optical notch at 480 nm and 620 nm). Being a crossover study, carry-over effects were analyzed, and even though the result was inconclusive, judged by the trend and moderate outliers, the group decided that, at most, there was a slight suggestion of a carry-over effect during the washout phase. The group also observed that the number of days with photophobia declined significantly when wearing the 620 nm lenses and not when wearing the 480 nm lenses, which led them to state that the bi-stable nature of melanopsin had a role in this experiment. They concluded that wearing these specially designed spectacles may be of help in chronic migraine.

Yuhas et al. studied the hypothesis that ipRGCs may have a persisting depolarization of resting membrane potential, thus becoming more sensitive to light in traumatic brain injury (TBI) patients [11]. This disruption in brain function has long-term sequelae that frequently include photophobia, which can persist for months [89,90], as well as accommodative dysfunction, oculomotor deficit, and visual defects, to name a few [91,92,93,94]. Using red light (625 nm) as a control and investigating the effects of blue light (470 nm), the group concluded that the relative contribution of ipRGCs was not altered in the mild TBI-associated photophobia group when compared to the controls. They did, however, observe a greater variability regarding pupil responses to blue light in the case participants’ data which, together with difficulty in sleep initiation and maintenance or excessive sleepiness that some TBI subjects exhibited, was interpreted as an ipRGC dysfunction.

Relying on the idea that there is a positive correlation between the stimulated retinal area and melanopsin activity [95,96], Zivcevska et al. [52] postulated that photophobia may be a perceptually summated experience. The authors wanted to evaluate whether, by using the psychophysical method of constant stimuli to assess perceptual response and maintaining consistent retinal stimulation (achieved by using dilating eye drops—Phenylephrine 2.5%), there would be a difference in “visual discomfort” in visually normal subjects in monocular as well as binocular viewing conditions. The experiment was conducted considering the properties of the ipRGCs, using blue (470 nm) and red (635 nm) stimuli. The group’s results, consistent with other studies, reinforce the idea that visually normal subjects are less sensitive to red light stimulation, with higher thresholds of sensitivity, stressing the idea that blue light can generate visual discomfort at much lower levels of intensity than other forms of light [30,32,52,97]. Eliminating habituation and anticipation errors by using the method of constant stimuli and a randomized sequence of blue and red-light intensities, the authors quantified this perceptual phenomenon and developed a new psychometric test that would bring its benefits in the current new working frame where there is more knowledge about ipRGCs than previously. They concluded that binocular viewing conditions have a lower threshold of discomfort, especially when paired with blue stimuli of a wavelength that correlates with the peak spectral intensity of melanopsin [13]. The group is the first to show a wavelength-dependent perceptual difference. This study adds new data that further support the involvement of the melanopsin pathway in light sensitivity perception and is the first to use pharmacological mydriasis in its experimental protocol to control for variation in retinal exposure both within and between test trials.

Migraineurs with visual aura seldom experience comfortable lighting conditions and as such resort to using tinted lenses. It has been shown by Aldrich et al. [98] that normal subjects, when faced with a choice of selecting the most comfortable color of light for viewing text, will select those that are similar to the ones provided by daylight and artificial lighting—i.e., close to the Planckian locus in comparison to patients with migraine with visual aura who tend to choose colors away from it—i.e., more saturated. Vieira et al. assessed the impact on the visual performance of such colors and observed that, when subjects were given glasses with a chromaticity that they had selected beforehand as being comfortable for reading text, the visual search performance of these subjects improved, especially in the migraine with aura study group [53]. When examined more closely, it became clear that the color of the lenses was not chosen solely based on the blockage level of the energy transmitted to the ipRGCs, proving and reinforcing two ideas: 1) the mechanism of photophobia is a complex one that includes more than ipRGCs, and 2) deterred visual function can be normalized by colored lenses, strengthening the color–migraine–light discomfort link.

The study conducted by Kaiser et al. [54] found a clear dissociation between explicit and implicit measures of photophobia in subjects experiencing interictal light discomfort. The authors used the method of silent substitution stimulation to target melanopsin, cones, or both while the central 5° of the visual field was blocked to minimize the effects of macular pigment. By using OO-EMG and an infrared camera to quantify blinking, the group observed that the increased stimulus contrast was linked to greater discomfort from light. This was true for both migraine study populations. In particular, patients with visual aura migraine exhibited an enhanced OO-EMG response to light at a 400% contrast response, but not at 200% or 100%. It was found that both cones and ipRGCs produced larger OO-EMG responses regarding tonic (squinting) and phasic (blinking) eye closure in the study group with visual aura migraine when compared to the rest. Thus, the study adds to a narrow dataset that stresses the differences in migraines between subjects with and without visual aura. As a response to bright light, people engage in both blinking and squinting activities. The photic blink reflex closes in the brainstem [99] and is intimately linked with the acoustic and tactile blink reflex [100]. Given the dissociation between explicit and implicit discomfort, it can be assumed that the physiological mechanism of the reflexive eye closure takes another path at some point from that of an enhanced conscious report of visual discomfort, as studies show a diverse and widespread projection of ipRGCs to both cortical (visual and somatosensory thalamic nuclei) and subcortical (the brainstem) sites. The location in the brainstem is more suspected than demonstrated, as to our knowledge, no neuroanatomic site has been described with these properties.

Zele et al. studied the contribution of ipRGC pathways to photophobia using different settings of light and different action spectra models—single, binary, and tertiary analytical models. Thresholds were estimated using both verbal reports and EMG of participants exposed to different wavelengths (461 nm, 525 nm, 630 nm), and the melanopsin function was quantified using pupillometry. The pupillometry data regarded the pupil light reflexes (PLR) and the post-illumination pupil response (PIPR) to a narrowband set of 1 s stimuli pulses—461 nm or 630 nm—to assess both cone and melanopsin pathways. Rod action spectrum was not included in this model as it did not significantly improve it. If involved in photophobia, the role of rods is likely mediated by luminance and melanopsin pathways [101]. In both study groups, it was observed that objective EMG photophobia thresholds were wavelength dependent, with the one for the red stimuli being higher than the one for the blue or green stimuli. As expected, the PLR metrics were normal in both of Zele et al.’s study groups, but inner retinal hyperexcitability was shown by the migraineurs group. They displayed a supranormal and prolonged PIPR, suggestive of an ipRGC hyperfunction. The binary action spectra, melanopsin and cones, best described the retinal pathway mediating photophobia in both healthy controls and migraineurs, with ~1.5× higher melanopsin weighting than cone luminance weighting. Hypersensitivity of the melanopsin pathway was observed and a combined path in a higher cortical center of the retino-geniculate, and a magnocellular pathway was proposed. As such, the cone luminance pathway was identified by Zele et al. as being a second retino-cortical pathway subserving photophobia [56]. 

In another study, Ali et al. [55] observed the effects of ipRGC stimulation on migraine severity. In their randomized, open-label, crossover, single-site study, one of the three objectives was to assess whether testing migraineurs with blue mfPOP stimuli would enhance the symptoms when compared with yellow stimuli. Migraine sufferers were not divided into groups based on the presence of visual aura. The group chose a slow blue protocol (1 s duration of stimulus) to evaluate the responses that are mainly driven by melanopsin as seen in other studies [102,103,104] and a fast yellow protocol (33 ms) for the evaluation of the cone luminance pathway. Regardless of the protocol, the number of subjects experiencing a migraine attack during the first day and the first three days remained non-significant, and it did not alter the number of migraine episodes per week. As a second objective, the group evaluated the ability of the blue and yellow stimuli protocols to detect pupillary responses in the period following a migraine attack but found no predictive association. Concerning their last aim, to see if mfPOP could be a future diagnostic tool, the group found only modest changes that were not global but rather five or more regions of asymmetry between eyes per field, with about five inferior temporal regions of the yellow protocol fields displaying consistent defects. These results are consistent with the studies of McKendrick et al. [105,106] and are unlikely to be due to a generalized autonomic dysfunction of the pupil. A meta-analysis performed by Feng et al. [107] showed that the retinal fiber layer (RNFL) was most thinned superior-nasally, which corresponds to such defects. Furthermore, consistent with the work of Cambron et al. [108], there were no differences found regarding pupil metrics. Even though they were the first to study migraine occurrence using a stimulus specifically designed for ipRGC stimulation (blue color and a stimulus duration of 1000 ms), discarding other characteristics such as stripe or check patterns [109], spatial frequency above two cycles/degree, and stimuli being delivered synchronously across the whole retina made it hard to trigger such an event, which in turn forces one to interpret these results with caution.

#### 4.3.2. Suppression of Vagal Activity Associated with Autonomic and Psychomotor Arousal

Given their relationship with various structures in the brain, such as the lateral geniculate nucleus (LGN), the suprachiasmatic nucleus (SCN), and the OPN, it is only reasonable to assume that the ipRGCs have an abundance of effects (such as melatonin secretion, alertness [18,80,81,110,111]), the most recently studied ones being on heart rate and heart rate variability. 

Colored lights and white lights are often selected for their aesthetic effects, but the human body’s response to the melanopsin-stimulating component emitted by these lights is hardly ever taken into consideration. This is of particular importance, especially for those that have a high amount of melanopsin-stimulating photon flux density (MSPFD). The cardiac autonomic function has been shown to be modified by such lights [50]. Using organic light-emitting diode (OLED) lights, Yuda et al. [50] first determined the impact of red (chromaticity (x = 0.63, y = 0.34), 39 lx), green (chromaticity (x = 0.33, y = 0.62), 71 lx), and blue (chromaticity (x = 0.14, y = 0.16), 10 lx) and different illuminance levels of blue (chromaticity (x = 0.14, y = 0.16), 10 lx, 5 lx, 2 lx) on the cardiac autonomic function. The HF power was used as an index of vagal cardiac modulation for cardiopulmonary resting [50,112,113,114]. With subjects synchronizing their breathing to a specific frequency (0.25 Hz), the authors showed that, before and during lighting, there was a significant decrease in HF and an increase in the LF-to-HF ratio (LF/HF). This difference was greater for blue light than for red and green light. The same result was obtained when comparing the before and after lighting periods, but with no regard to color. With participants still maintaining a paced breathing pattern, their second experiment showed that only the highest illuminance level (blue light 10 lx vs. 5 lx vs. 2 lx) induced significant changes in heart rate and heart rate variability, in light and darkness following light periods (suggesting a prolonged effect). Using a paced breathing protocol, these results may reflect the effects of colored lights on central vagal function, thus eliminating the bias effect of respiration frequency on heart variability. 

Yuda et al. further studied the impact on the autonomic and psychomotor arousal level of the melanopic component (absolute and relative amount) of blue light (chromaticity [x = 0.14, y = 0.16], 13 lx, MSPFD 0.23 μmol/m^2^/s, Relative MSPFD 72%) green light—lesser amount of relative MSPFD (chromaticity [x = 0.33, y = 0.62], 91 lx, MSPFD 0.14 μmol/m^2^/s, Relative MSPFD 17%) and white light—greater absolute amount but a lesser relative MSPFD (chromaticity [x = 0.44, y = 0.41], 158 lx, MSPFD 0.38 μmol/m^2^/s, Relative MSPFD 14%) [51]. The authors showed significant effects of light color and session on heart rate, HF power, and LF power. Blue light again had a higher impact in decreasing HF power and increasing heart rate than either green or white light. It also demonstrated better arousal effects than green light, but there was no difference when compared to white light. Yuda et al. showed that the difference in absolute melanopic component plays an important role as suggested by the difference of effects between blue and green OLED [50,51], and by the relative component shown by the differences between blue and white light, although results should be interpreted cautiously. The pupil plays a significant part in the number of photons that reach the retina and by default in the MSPFD. Yuda et al. studied this aspect, but the number of subjects included was low (n = 7) and the subjects were different from the main study, making it hard to form a firm conclusion.

Because of the small number of participants, it is hard to apply these results to the general population. Studying the consequences of blue light exposure at a systemic level may also require longer study periods than those used in Yuda et al.’s studies [50,51,115]. This eliminates the possibility of a situation in which the systemic response to blue light may have not been fully saturated and may have progressed further than the study period. Moreover, it is unclear whether the washout period between stimulation was enough or should have been prolonged. Because, in some studies, the light intensity of illumination was not standardized, the possible effect of intensity cannot be entirely excluded. However, by having enrolled healthy participants in the studies, the results highlight the effects of blue light on our everyday life. Even though studies have shown that blue light induces a sustained higher arousal and alertness state, these results should be interpreted with a great deal of caution, as baseline data was not collected and possible differences in autonomic (some also influenced by the menstrual cycle) and psychomotor performance cannot be firmly excluded. Starting from the idea that white light exposure to the inferior retina causes greater melatonin suppression when compared with exposure to the superior retina [116,117], Yuda et al. investigated the autonomic effects of blue light emitted by OLEDs (chromaticity (x = 0.14, y = 0.16), 15.4 lx, MSPFD of 0.28 μmol/m^2^/s, relative MSPFD 75%) [115] and found no significant exposure angle-dependent effect on the autonomic indices of HRV. 

## 5. Future Directions

Few research groups have used dilating drops. Having a constant retinal zone stimulated and assessed makes future studies less heterogeneous, their data easier to interpret at a global level, and clearer conclusions to be drawn. The level of each participant’s MPOD is important for future studies to consider because of its role in photophobia and oxidative stress. Subjective methods of assessing the impact of blue light on the retina are found in most studies. Even though the use of questionnaires is a cost-effective method compared to a more objective one—OCT, ERG, mfERG, CFF—we consider that future studies should make much more use of the latter ones because they are more predictable, reliable, measurable, and precise.

We are also of the opinion that, by testing patients in interictal periods, it is difficult to detect differences that may be attributed to changes occurring during the migraine attack. Different therapeutical approaches have been adopted to treat migraine and its accompanying symptoms, from tinted lenses to pharmacological manipulation, all of them stressing the ever-important role that the duo blue light-ipRGC plays in migraine pathophysiology. Safety and efficacy are both important when considering new lighting sources, and despite the limitations of these studies, the research conducted on human participants so far represents the first steps into the developing area of luminaires, where LEDs and OLEDs, and implicitly blue light and MSPFD, play such an important part in the contemporary life environment—in the workplace, at home, and in healthcare environments. Care should be taken as cases of subjects experiencing migraine during or after exposure to silent substitution stimuli have been reported [55].

The problem of the bi-stable state of melanopsin raised by Hoggan et al. [49] opens a whole new dimension for optical notch filters—should they also be centered around 590 nm, the wavelength that converts melanopsin from its inactive to active state, thereby limiting the number of molecules capable of reacting to blue light? Another interesting future direction for studies is to explore individual differences between migraine sufferers with and without aura, as this is still a subject of debate.

## 6. Conclusions

In previous studies, there is little evidence that the theoretical protection of different blue-blocking filters against retinal degeneration is clinically significant. To date, many findings are theoretical or based on observations in cell culture or animal experiments. As far as subject enrolling and testing are concerned, the small number of participants makes it hard to generalize results and derive a firm conclusion. Few studies have adequate patient follow-up periods to observe both the damaging effects of blue light and the protective effects of blue light-filtering ophthalmic devices on the human retina. After having read the literature regarding experimental designs, we observed that research groups defined the mesopic setting differently, ranging from 0.1 lx to 50 lx, and this inconsistency only makes such obtained data on an already-controversial topic even harder to interpret. The wavelength and stimulus length used for experimental ipRGC exposure also vary between studies.

We find it difficult to express with certainty that blue light exerts a harmful effect at the level of the human retina, but with more and more research groups exploring this topic, using designs that resemble today’s way of living and enrolling healthy young participants who use VDTs often, studies are starting to obtain more relevant and common results that all point towards the possible noxious effect of long-term blue light exposure. Despite having the groundwork laid for these experiments more than two decades ago, questions remain about the effects of this short-wavelength light on the human retina. We hope that the present narrative review will provide useful insights for future studies.

## Figures and Tables

**Figure 1 ijms-24-05998-f001:**
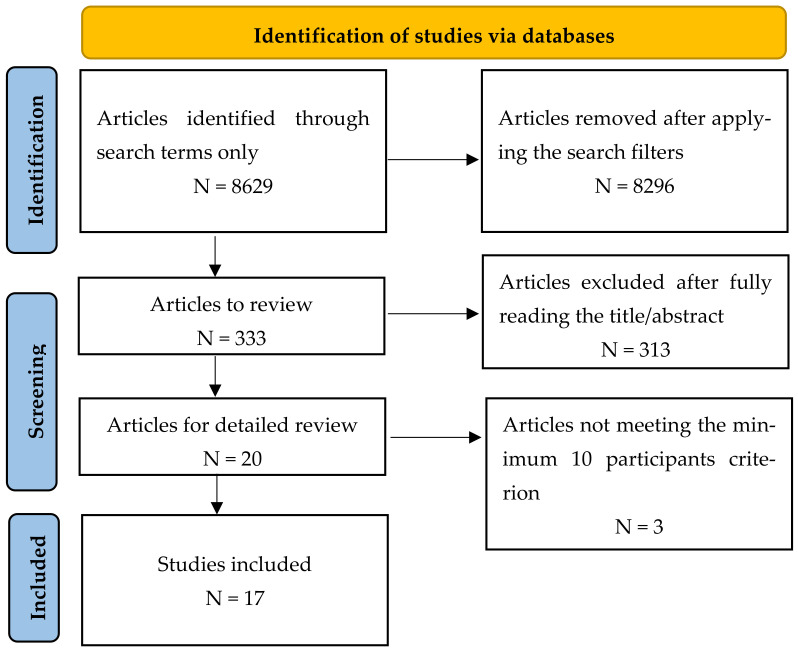
Flow chart of the selection of reviewed articles.

**Table 1 ijms-24-05998-t001:** Studies focusing on the protective effects of yellow-tinted IOLs upon the retina.

Year	Title	Authors	Subjects (Eyes with Yellow/Clear IOL)	Element of Study	Primary Endpoint	Type of Study	Maximum Follow-Up	Outcome
2010	Comparison of visual performance with blue light-filtering and ultraviolet light-filtering intraocular lenses	Neumaier-Ammerer et al. [41]	76 (37/39)	Cones and rods	VA, CS, Color vision	Prospective, randomized, double-blind control trial	2 months	Negative: no difference between tested parameters (except for color vision)
2011	Effects of blue light-filtering intraocular lenses on the macula, contrast sensitivity, and color vision after a long-term follow-up	Kara-Junior et al. [42]	25 (25/25)	Cones and rods	CS, Color vision, OCT Ophthalmoscopy	Prospective randomized control trial	5 years	Negative: no differences between tested parameters
2015	Prevention of increased abnormal fundus autofluorescence with blue light-filtering intraocular lenses	Nagai et al. [43]	131 (52/79)	RPE cells	FAF	Prospective comparative observational study	2 years	Positive: Lower FAF abnormalities and AMD incidence in the yellow-tinted IOL group
2015	Color of Intra-Ocular Lens and Cataract Type Are Prognostic Determinants of Health Indices After Visual and Photoreceptive Restoration by Surgery	Ayaki et al. [44]	206 (135/71)	Cones and rods	Japanese versions of: VFQ-25 PSQI	Prospective comparative observational study	7 months	Positive: improvement of VFQ-25 (yellow-tinted IOL) and PSQI (clear IOL)
2016	Effect of blue light-filtering intraocular lens on color vision in patients with macular diseases after vitrectomy	Mokuno et al. [45]	67 (27/40)	Cones and rods	Color vision	Prospective comparative observational study	16 months	Negative: no differences between study groups

VA = visual acuity; CS = contrast sensitivity; OCT = optical coherence tomography; FAF = fundus autofluorescence; VFQ-25 = National Eye Institute Visual Function Questionnaire; PSQI = Pittsburg Sleep Quality Index.

**Table 2 ijms-24-05998-t002:** Studies focusing on the electrophysiology of the retina.

Year	Title	Authors	Subjects	Element of Study	Primary Endpoint	Type of Study	Outcome
2011	Impact of blue vs. red light on retinal response of patients with seasonal affective disorder and healthy controls	Gagné et al. [46]	20	Cones and rods	ERG	Prospective controlled study	Positive: blue light decreases maximal ERG response
2018	Effect of Heat-Killed Lactobacillus paracasei KW3110 Ingestion on Ocular Disorders Caused by Visual Display Terminal (VDT) Loads: A Randomized, Double-Blind, Placebo-Controlled Parallel-Group Study	Morita et al. [47]	62	Cones and rods	CFF, VAS, CS	Prospective, randomized, controlled study	Neutral: significant improvement 4 weeks after intake, but not during the 8th week
2021	Blue Light from Cell Phones Can Cause Chronic Retinal Light Injury: The Evidence from a Clinical Observational Study and a SD Rat Model	Li et al. [48]	25	Cones and rods	mfERG	Prospective, randomized, controlled study	Positive: reduced amplitude of parafoveal regions

ERG = electroretinogram; CFF = critical flicker frequency; VAS = visual analog scale; mfERG = multifocal ERG.

**Table 3 ijms-24-05998-t003:** Studies focusing on the systemic effects of blue light.

Year	Title	Authors	Subjects	Element of Study	Primary Endpoint	Type of Study	Outcome
2016	Thin-film optical notch filter spectacle coatings for the treatment of migraine and photophobia	Hoggan et al. [49]	48	ipRGC	Headache impact using HIT-6	Randomized, double-masked, crossover	Positive: Reduction in HIT-6 score + Unexpected result for the 620 nm lenses
2016	Suppression of vagal cardiac modulation by blue light in healthy subjects	Yuda et al. [50]	12	ipRGC	HRV indices—HF, LF/HF	Open-label, Non-Controlled Trial	Positive: Lower HF (greatest with blue light) and increased LF/HF
2017	Enhancement of autonomic and psychomotor arousal by exposures to blue wavelength light: importance of both absolute and relative contents of melanopic component	Yuda et al. [51]	10	ipRGC	Heart rate, HRV indices (HF, LF/HF) + performance of PVT	Open-label, Non-Controlled Trial	Positive: lower heart rate, HF, and reaction time, but no difference in LF/HF
2017	Blue and Red Light-Evoked Pupil Responses in Photophobic Subjects with TBI	Yuhas et al. [11]	36	ipRGC	Pupil fluctuation	Open-label, Non-Controlled Trial	Negative: no differences between groups regarding blue light
2018	A Novel Visual Psychometric Test for Light-Induced Discomfort Using Red and Blue Light Stimuli Under Binocular and Monocular Viewing Conditions	Zivcevska et al. [52]	11	ipRGC	Light discomfort thresholds	Open-label, Non-Controlled Trial	Positive: greater discomfort for blue light under both monocular and binocular stimulation
2020	Preference for Lighting Chromaticity in Migraine with Aura	Vieira et al. [53]	54	ipRGC	Visual search task	Cross-sectional laboratory study	Positive: visual search time decreased
2021	Reflexive Eye Closure in Response to Cone and Melanopsin Stimulation: A Study of Implicit Measures of Light Sensitivity in Migraine	Kaiser et al. [54]	60	ipRGC	OO-EMG, VDS	Non-Randomized Controlled Trial	Positive: greater OO-EMG activity and visual discomfort for migraineurs
2021	Assessing migraine patients with multifocal pupillographic objective perimetry	Ali et al. [55]	62	ipRGC	Migraine headache diary, mfPOP	Randomized, open-label, crossover	Negative: no differences between used protocols and study groups for the first two aims. Moderate changes for the yellow protocol about their third aim
2021	Melanopsin hypersensitivity dominates interictal photophobia in migraine	Zele et al. [56]	23	ipRGC	EMG, Pupillometry	Non-Randomized Controlled Trial	Positive: lower EMG thresholds and higher PIPR in blue and green light settings

HRV = heart rate variability; HF = high frequency; LF = low frequency; PVT = psychomotor vigilance test; OO-EMG = orbicularis oculi electromyography; VDS = visual discomfort score; mfPOP = multifocal pupillographic objective perimetry; PIPR = post-illumination pupil response.

## Data Availability

The datasets used and analyzed during the current study are available from the corresponding author upon reasonable request.

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
