# Peer review of "Blue Light—Ocular and Systemic Damaging Effects: A Narrative Review"

_ijms, 2023, doi:10.3390/ijms24065998_

Round 1

Reviewer 1 Report

In this review article entitled “Blue light and the retina – local and systemic damaging effects: a narrative review”, Authors want to summarize knowledge about blue light exposure and outcome on people’s health. The subject is of potential interest. However, some issues need to be addressed before accepting this work for publication.

Main concern:

- In the first paragraph of the introduction, a proper description of the retina structure and its different layers would be helpful, with the addition of a schematic of the retina.

-A schematic of the ipRGC and their brain targets (and the functions of these brain nuclei) would also help the readers.

- the cross talk between the ipRGC and the trigeminal nerve is not very clear. For a broad audience, this point should be better explained.

- It is not clear how the authors decided to select 20 articles from the 333 studies that go through the initial screening.

-As ipRGC main brain target is the suprachiasmatic nucleus, it was expected to discuss the effect of blue light on circadian rhythms that control many physiological functions (body temperature, hormones release, sleep..)

-lines 231-237: this paragraph is not clear.

Minor points:

-          There are some typos, for example line125 (replace “of” with “on”), line 212 (the paragraph title)…

-          Authors should explain from line 223 what VTD stand for (the explanation is given only in line 242)

-          Please replace RGCs by RGC (otherwise it would be double “s” at the end of cells)

Author Response

Dear Reviewer, we have improved our manuscript, taking into consideration your observations

- In the first paragraph of the introduction, a proper description of the retina structure and its different layers would be helpful, with the addition of a schematic of the retina.

Answer:This part has been expanded for a better understanding of the subject.

-A schematic of the ipRGC and their brain targets (and the functions of these brain nuclei) would also help the readers.

A: The discussion regarding ipRGCs has been expanded, but the discussion of all their targets and brain nuclei would derail from the article’s main subject, which is the effects of blue light. Numerous reviews regarding the connections of ipRGCs exist, some also in the Reference section.

- the cross talk between the ipRGC and the trigeminal nerve is not very clear. For a broad audience, this point should be better explained.

A: This discussion has been expanded for a better understanding of the subject.

- It is not clear how the authors decided to select 20 articles from the 333 studies that go through the initial screening.

A: The Results section has been modified to make it clearer

-As ipRGC main brain target is the suprachiasmatic nucleus, it was expected to discuss the effect of blue light on circadian rhythms that control many physiological functions (body temperature, hormones release, sleep..)

A:The article’s title states that only the damaging effects are taken into consideration. It is explicitly stated that these functions will not be discussed in the present manuscript. The phrase regarding the aim of the manuscript has been modified to avoid this possible confusion: “Thus, the present narrative review aims to offer a wider perspective of the harmful biological consequences after blue light exposure of the human retina, gathering and interpreting the effects both at a local and systemic level. Effects regarding photoentrainment, melatonin and other hormone secretion, sleep quality and next morning alertness, axial length, and refractive status of the eye concerning blue light have not been taken into consideration.” 

-lines 231-237: this paragraph is not clear.

 A: Rephrased to make it clearer.

We have made the minor corrections. We kept however the singular and plural of ipRGC (intrinsically photosensitive retinal ganglion cell)/ ipRGCs. We have rephrased to avoid confusion

Reviewer 2 Report

This is an excellent review of the basic science and visual function data available to date regarding blue light exposure on the retina.

Author Response

Thank you!

Reviewer 3 Report

Antemie et al. present a narrative review of ocular and systemic effects of blue light. Thanks for the opportunity to review, as the manuscript is very interesting, timely, relevant, well-reference, ethically prepared, and applicable. While the manuscript requires extensive English editing, it is reasonably well-written. Interestingly, while the readability is marginal at the beginning, it seems to improve as you get deeper/further into the work.

The idea of a narrative review of this subject (blue light effects on the retina, lens, and some body systems) is timely, but the manuscript is somewhat disorganized. In addition, if the editors elect to publish (an outcome I would ultimately favor), I have offered up below many suggestions to improve organization, readability, and understanding.

1. Title needs changed. It currently reads as if there could be systemic effects on the retina(?). Suggest: "Blue light - ocular and systemic damaging effects: a narrative review"

2. Lines 10-13: Suggest this edit: "Blue light has always existed in nature, as. Light is a fundamental aspect of our life, being involved in the regulation of numerous processes in our body. While blue light has always existed in nature, Given nowadays modern way of living, with an the ever-growing number of electronic devices that make use of this short wavelength (blue) light, the human retina has seen an increased exposure to it."

3. Lines 19-21: Suggest: "The current manuscript aims to gather the most recent relevant data, reveal encountered pitfalls, and suggest future directions for studies regarding local and or systemic effects of blue light retinal exposures."

4. Line 42: Suggest: "...that the retinoid A2E (N-...."

5. Lines 44-47: Suggest: "There are many recent attempts to block blue light phototoxicity, lately,  ranging from spectacle lenses to intraocular lenses (IOLs) and even a dietary approach; all with variable and controversial benefits [5,10-12]."

6. Lines 66-68: Suggest: "Many theories have been proposed along the time,
starting with Hopkinson’s pupillary hippus theory [25] that who imagined the iris as having a role in the pain-signaling mechanism for the visual discomfort."

7. Lines 76-77: Suggest: "At least one group (Stringham et al.; [31]) determined that by decreasing wavelength this process increased photophobia. They also showed that the macular pigment (MP) has an important role in..."

8. Lines 83-89: Suggest: "Over the past decade, a growing number of studies have centered their focus on the effects of blue light on the retina and--more recently--on the human body as a whole. This is especially relevant given the nowadays context – a high use of smartphones and tablet devices [33-36] that use a solid-state lightning technology which emits a higher amount of blue light despite its appearance to emit white light appearance [35,36]. Blue light plays a significant role in our day-to-day lives, penetrating so deeply into our education, modern industries that run 24/7, and--most importantly--into our lifestyle, that it seems impractical and almost impossible to avoid exposure to it."

Lines 97-100: Suggest: "This manuscript was designed prepared following the indications and procedures that were provided by Green at al [37], Grant et al [38], and Pautasso M [39], as well as following the guidelines described by Baethge et al [40] when speaking about regarding the SANRA (Scale for the Assessment of Narrative Review Articles)."

Line 103: The authors used "intrinsically photosensitive retinal ganglion cell" as a search term. It would be useful to show in a flowchart how/why articles were removed from consideration. For example, Neumaier et al. [41], Kara-Junior et al. (50], etc. did not mention ipRGCs (only rods and cones). I suggest the following: For Results section, please rework lines 107-110 to show FIRST FILTER --> 333 abstracts. NEXT FILTER --> 20 abstracts. AFTER FULLY READING --> 17 studies included. Be very specific about the filters and how/why articles excluded. Then please include reference numbers  next to Authors in Table 1. For example: "Neumaier-Ammerer B et al. [41]"

Table 1: Do the numbers in parentheses represent men/women? If so, please indicate in legend or other.

Line 124: Please consider changing "In the past" to "Previously" here. Also, please make this change everywhere in the manuscript.

Line 132: Please consider changing "To this day" to "Currently," Also, please make this type of change throughout mansucript.

Line 170: "...its changes 2 years..." should read "...its changes two years..." (While MDPI may not have this standardized, numbers < 10 should be spelled out. Please consider this change throughout the manuscript.)

Line 206: "...lightning..." should be changed to "...lighting..." The entire manuscript needs to undergo careful English editing. It it readable but needs a lot of small changes.

Line 279: The lenses are FL-41 tinted spectacles (not FL-14).

Line 471-472: Please change to "...OLED (chromaticity [x=0.14,y=0.16], 15.4 lx, MSPFD of 0.28 mmol/m2/s, relative MSPFD of 75%)[115].

I have offered many suggestions of grammar/English edits. Please consider all these, but there are many more that need completed. Again, the manuscript would benefit greatly from an extensive English edit. However, it is interesting and timely. I would be glad to review any re-writes if editorial team moves forward with revisions.

Author Response

  1. Title needs changed. It currently reads as if there could be systemic effects on the retina(?). Suggest: "Blue light - ocular and systemic damaging effects: a narrative review"

Changed as proposed.

  1. Lines 10-13: Suggest this edit: "Blue light has always existed in nature, as. Light is a fundamental aspect of our life, being involved in the regulation of numerous processes in our body. While blue light has always existed in nature, Given nowadays modern way of living, with anthe ever-growing number of electronic devices that make use of this short wavelength (blue) light, the human retina has seen an increased exposure to it."

Changed as proposed.

  1. Lines 19-21: Suggest: "The current manuscript aims to gather the most recent relevant data, reveal encountered pitfalls, and suggest future directions for studies regarding local and or systemic effects of blue light retinal exposures."

Changed as proposed.

  1. Line 42: Suggest: "...that the retinoidA2E (N-...."

Changed as proposed. [desi apare si mai jos ca e retinoid…]

  1. Lines 44-47: Suggest: "There are many recentattempts to block blue light phototoxicity, lately, ranging from spectacle lenses to intraocular lenses (IOLs) and even a dietary approach; all with variable and controversial benefits [5,10-12]."

Changed as proposed

  1. Lines 66-68: Suggest: "Many theories have been proposed along the time,
    starting with Hopkinson’s pupillary hippus theory [25] thatwho imagined the iris as having a role in the pain-signaling mechanism for the visual discomfort."

Changed as proposed

  1. Lines 76-77: Suggest: "At least one group (Stringham et al.; [31]) determined that by decreasing wavelength this process increased photophobia. They also showed that themacular pigment (MP) has an important role in..."

Changed as proposed

  1. Lines 83-89: Suggest: "Over the past decade,a growing number of studies have centered their focus on the effects of blue light on the retina and--more recently--on the human body as a whole. This is especially relevant given thenowadays context – a high use of smartphones and tablet devices [33-36] that use a solid-state lightning technology which emits a higher amount of blue light despite its appearance to emit white light appearance [35,36]. Blue light plays a significant role in our day-to-day lives, penetrating so deeply into our education, modern industries that run 24/7, and--most importantly--into our lifestyle, that it seems impractical and almost impossible to avoid exposure to it."

Changed as proposed [nu stiu la ce se refera asta: “and--more recently--on”]

Lines 97-100: Suggest: "This manuscript was designed prepared following the indications and procedures that were provided by Green at al [37], Grant et al [38], and Pautasso M [39], as well as following the guidelines described by Baethge et al [40] when speaking about regarding the SANRA (Scale for the Assessment of Narrative Review Articles)."

Changed as proposed

Line 103: The authors used "intrinsically photosensitive retinal ganglion cell" as a search term. It would be useful to show in a flowchart how/why articles were removed from consideration. For example, Neumaier et al. [41], Kara-Junior et al. (50], etc. did not mention ipRGCs (only rods and cones). I suggest the following: For Results section, please rework lines 107-110 to show FIRST FILTER --> 333 abstracts. NEXT FILTER --> 20 abstracts. AFTER FULLY READING --> 17 studies included. Be very specific about the filters and how/why articles excluded. Then please include reference numbers  next to Authors in Table 1. For example: "Neumaier-Ammerer B et al. [41]"

The first part of the Results section has been rephrased to be clearer about the selection of the final 17 articles, making the use of a flowchart unnecessary. Reference numbers have been added next to the Authors in Table 1.

Table 1: Do the numbers in parentheses represent men/women? If so, please indicate in legend or other.

The numbers in parentheses represent the number of eyes that received a yellow/clear IOL, as it is written in the parentheses in the table header. [si aici a fost pe langa] 

Line 124: Please consider changing "In the past" to "Previously" here. Also, please make this change everywhere in the manuscript.

Changed as indicated.

Line 132: Please consider changing "To this day" to "Currently," Also, please make this type of change throughout mansucript.

Changed as indicated.

Line 170: "...its changes 2 years..." should read "...its changes two years..." (While MDPI may not have this standardized, numbers < 10 should be spelled out. Please consider this change throughout the manuscript.)

Changed as indicated.

Line 206: "...lightning..." should be changed to "...lighting..." The entire manuscript needs to undergo careful English editing. It it readable but needs a lot of small changes.

Changed as indicated.

Line 279: The lenses are FL-41 tinted spectacles (not FL-14).

Change as indicated

Line 471-472: Please change to "...OLED (chromaticity [x=0.14,y=0.16], 15.4 lx, MSPFD of 0.28 mmol/m2/s, relative MSPFD of 75%)[115].

The change regarding parentheses has been made, but regarding the measurement unit for MSPFD it is μmol/m2/s, and not mmol/m2/s, the latter being 1000x greater.

I have offered many suggestions of grammar/English edits. Please consider all these, but there are many more that need completed. Again, the manuscript would benefit greatly from an extensive English edit. However, it is interesting and timely. I would be glad to review any re-writes if editorial team moves forward with revisions.

We have English-proofed the article, thank you for your observations

Reviewer 4 Report

It is interesting paper which widely search for the effect of blue light exposure on human.

Minor changes

Line 212: tu à to

Line 825: 118. delete

Author Response

Thank you!

We have made the changes

Round 2

Reviewer 1 Report

The author revised their manuscript according to the comments. It is ready to be published.